# Specific Needs of Patients and Staff Reflected in the Design of an Orthopaedic and Rehabilitation Hospital—Design Recommendations Based on a Case Study (Poland)

**DOI:** 10.3390/ijerph192215388

**Published:** 2022-11-21

**Authors:** Agata Gawlak, Marta Stankiewicz

**Affiliations:** Faculty of Architecture, Institute of Architecture, Urban Planning and Heritage Protection, Poznan University of Technology, 60965 Poznan, Poland

**Keywords:** hospital design, patient, staff, architecture, orthopaedic hospital, rehabilitation, participation

## Abstract

This article presents results of the research conducted with the use of participatory methods by the Faculty of Architecture, Poznan University of Technology on architectural design of healthcare facilities. The studies concerned the needs of patients and hospital staff in an orthopaedic and rehabilitation hospital. Preferences and expectations of all the users of healthcare facilities should be considered as early as at the stage of planning and designing. The hospital profile and the type of its users predetermine the architectural design in the area of the building functions, its internal circulation and interior design. Participation of the user in the process of compiling design recommendations is a chance for a designer to confront the original assumptions with expectations and to adjust relevant solutions to factual needs of the users. This study, undertaken in a regional setting, provided an in-depth exploration of staff’s experiences of hospital space to indicate possible spatial improvements. Methods: The research was conducted on the basis of a case study of a renowned regional orthopaedic and rehabilitation hospital in Poznan, Poland. Rapid assessment methods and questions were examined to describe current approaches and synthesize results. Semi-structured interviews and thematic analysis identified staff and patient’s experiences. Result: Participation of hospital staff and patients resulted in design recommendations of high utility value. It was found that the two groups to a certain extent proposed similar recommendations; however, certain proposals submitted by the two groups were totally opposite. Conclusion: the research highlighted the importance of the active research methodology that engages the researcher/expert in the action and critical reflection process. Such a methodology can successfully underlie the formulation of accurate recommendations.

## 1. Introduction

### 1.1. Design of Healthcare Facilities and Its Social Impact

The aim of this research is to create project recommendations for orthopaedic hospital in Poland. The rank of participatory design that involves patients and staff increases if quantitative and qualitative research is combined. Functionally oriented, equipped with state-of-the-art technologies, determined with legislative requirements and financial constraints, architecture of healthcare facilities must remain sensitive and empathic towards their users by properly responding to their individual needs.

Hospitals are facing challenges inherent in high demand for medical services due to increased number of patients, as the proportion of patients over 65-years-old increases [1]. On the other hand, research shows that an individual approach to a patient in his/her recovery period plays an important role. This fact applies not only to the dedicated treatment procedures but also to hospital space adapted to individual needs of the building users [1].

Proper parameters of functional hospital layouts, reproduced as model solutions, can be viewed as an improvement that will not only contribute to the wellbeing of hospitalised patients but will also increase work efficiency of medical and non-medical staff [2]. The fact that efficiency of hospitals needs to be increased does not mean that it has to be undertaken at the expense of the customisation of medical services and recovery space to individual needs. Participation has become a method of integrated design [3]. In the overall recovery process, the quality of space and its customisation to individual needs of the patient is as important as the right diagnosis and dedicated therapy. As patients, we have become more and more demanding, we are no longer interested in standard solutions, we expect solutions that are tailored to our individual needs, requirements, lifestyle and personality. R. Ulrich’s Report [4]—a synthetic review of over 600 scientific publications on the impact of architectural solutions (physical environment) in hospital settings on the patient’s feeling of comfort and safety, the process of diagnosis, and therapy and on efficient work of staff—has been widely used by designers of hospital facilities. Arguments for adoption of a given methodology are first of all supported with the successful implementation of the evidence-based design in architectural designs of healthcare facilities. From the studies presented by Ulrich, it follows that well-being of patients may depend on various architectural factors and quality of space, where such quality is created at an early stage of a hospital or, speaking more broadly, of a healthcare facility design [5]. Moreover, hospital space may obtain basically different assessments depending on sociodemographic variables such as age, sex, waiting time for a medical appointment or hospital arrival hour. This confirms that respective adaptations of spatial features of built environment and climate characteristics related to, e.g., room temperature, air humidity, noise level, or light intensity largely affect the wellbeing of patients and should be taken into account at a stage of designing and organising hospital buildings in view of patients’ perceptive and adaptive capabilities. The literature distinguishes four basic aspects of the built environment as with regards to healthcare facilities that predetermine the level of patients’ satisfaction on the basis of their individual perception of space, i.e., the environment conditions, functional layout of a hospital, hygienic and sanitary conditions, and clear circulation system [1,2,3,4].

The above four criteria allow for a holistic assessment of space from the patient’s perspective. Nevertheless, sociodemographic factors as the patients’ gender, age, education that determine the respondents’ level of satisfaction should also be considered [6,7]. Research on the perception of hospital space by its users [8] conducted by a team of scientists from the UK (Monjur Mourshed and Yisong Zhao) shows that such factors as the lighting, furniture, and colour scheme do affect the way architectural space of a hospital is perceived by the patients. Based thereon, it has been proven that satisfaction of hospital users is determined by the quality of space. It has, moreover, been proven that incorporation of the users’ expectations into the hospital’s functional planning facilitates the creation of truly healing space [5,9]. It has been also confirmed that quality of architectural space of a hospital is created at a very early stage of designing and functional planning of the building. At the same time, another team from China and the USA (Tsai C.Y., Wang M.C., Liao W.T., Lu J.H., Sun P.H., Lin B.Y., Breen G.M.) pioneered studies on the level of satisfaction of hospital patients from medical services in the outpatient facilities [10], scale of their needs, requirements, and perception of space, accounting for sociodemographic factors. It turned out to have a significant impact on the assessment of functional and spatial solutions. Having surveyed 680 patients of 17 admission rooms, it has been found that [10]:Personal features of patients affect their satisfaction regarding hospital admission;”Face to face” opinion polls are more reliable;If patients’ expectations are taken into account at the modernisation of space of admission rooms, patients’ satisfaction level increases.

Having been able to analyse the results of the aforementioned research, the authors have adapted relevant research assumptions in order to transpose them onto Polish conditions. Thus, the authors have worked out their own questionnaire, accounting for those architectural aspects of admission rooms that the previous research found important for the wellbeing of patients and staff of an orthopaedic and rehabilitation hospital.

### 1.2. Situation of Healthcare Facilities in Poland in View of an Ageing Society

To meet the challenges of the 21st century, hospitals need to be transformed to properly address sociodemographic changes and an ageing society. As a result, we are facing higher demand for medical services that need to be provided 24 h a day/7 days a week to an increasing population of patients. Architectural designs of hospitals should be able to adequately respond to the changing needs of the users and this applies both to new and already existing buildings. We need to remember that not only medical technologies and treatment methods are subject to rapid changes. So are the preferences of the users, including their spatial needs [11].

Forecasts of how the healthcare system and residential architecture have to transform stress the importance of user participation in the overall process of architectural design of healthcare facilities [12], in this case a particular role of the patients in the treatment process and in the process of treatment management (patient-centred design and co-design). Architecture of a building directly affects building management, and thus indirectly affects the financial result of building management [13].

Costs of hospital treatment have been continuously rising as a result of the increasing number of seniors in the overall population (ageing society phenomenon) (Figure 1 and Figure 2).

Calculating globally, hospitalisation costs are mainly generated by patients aged 65+ (around 40%), this group is followed by patients aged from 45 to 65 (27%). The share in such costs by patients aged from 20 to 44 is 20% and patients aged under 20 generate the lowest hospitalisation costs (at a level under 10%) [14].

Patients aged 65+ generate the highest costs of medical treatment; moreover, a general share of that age group in the overall population will be rising, this without doubt will translate into increased hospitalisation costs in future [15].

According to the forecasts, in 34 years’ time public spending on medical services in European countries will double as compared to the current spending levels.

Hospitals can be designed to optimise building service fees, e.g., via optimisation of space and staff circulation paths or optimisation of the building fit. However, even the most efficient hospital management cannot prevent a financial crisis in future if the form and space of healthcare facilities is not revised adequately to the increasing demand for medical services with the reservation that it shall not undermine the quality of treatment and efficiency of the overall healthcare system [14], also attending to the quality of treatment and efficiency of the overall healthcare system [16].

According to the data from the Central Statistical Office of 2019, there are almost 1000 hospitals in Poland that provide their medical services in over 7000 departments with over 180,000 beds for patients [17].

Proper parameters of functional hospital layouts, reproduced as model solutions, could be viewed as an improvement that will not only positively affect the wellbeing of hospitalised patients but also will increase work efficiency of medical and non-medical staff [18].

At present, there is a clear tendency to restructuring and reducing the size of hospital departments due to the introduction of cutting edge treatment technologies that are not only more efficient but also improve the healing process and shorten the required hospitalization [19]. As far as Poland is concerned, the situation seems very favourable [16,17].

From the reports of the National Institute of Hygiene (National Institute of Public Health in Warsaw), it follows that Poland is a country in Europe with the highest number of beds per inhabitant (in 2019 there were about 600 beds per 100 thousand inhabitants—these figures are similar to those characteristic of such countries in Central Europe as Germany: 880 beds per 100 thousand inhabitants, Austria—767, Lithuania—743, Czechia—666. Taking such a favourable ratio in mind, it might seem that in this area the needs are fully satisfied, whereas in fact the number of beds does not represent the efficiency of the healthcare system and the operation of the hospital network. As an example, we can refer to the Scandinavian countries where the number of beds is very low but the quality of medical services is very high. In Norway, for instance, there are only 397 beds, in Sweden 261 beds and in Great Britain 280 beds per 100,000 inhabitants. The reason is that these countries have focused on outpatient treatment and more efficient prophylaxis [16].

What is interesting is that, at a high rate of beds in Poland and a common opinion on inefficiency of the healthcare system, the percentage of bed occupancy is relatively low and amounts to 68% (there are even hospitals with bed occupancy of 11%). Rates at the level of 80–85% are assumed to represent efficient use of hospital beds.

Denmark is a recommended model to follow with regards to the organisation of the healthcare system. The number of hospitals has been regularly decreasing there (in 2000 there were about 78 hospitals and in 2020 only about 20). Decisions on closing hospitals are made in response to demographic changes, shortages of staff, and above all, new organisational models and innovative technologies that allow us to shift the burden of healthcare from hospitals to outpatient facilities and places of residence [18]. Moreover, access to information facilitates better prophylaxis, e.g., via access to knowledge and educational activities.

In the years 2019–2020, we can observe that the number of beds has been slowly going down in Poland [16]. This is, however, mainly due to new legal provisions on the required number of employed nurses per one hospital bed. Increasing demand for professional nursing care is closely correlated with changing demographics. Forecasts assume higher spending levels in the healthcare sector by about 100% in the next 34 years. The control made by the Polish Supreme Chamber of Control upon an order from the Ministry of Heath shows that the declining number of hospital beds has failed to translate into the satisfaction of patients from the quality of medical services. On the contrary, their dissatisfaction continues to be very high. Yet, it can be observed that the group of satisfied patients grew by 4% in the inspected period (2008–2018), at the same time, the number of dissatisfied patients proportionately went down (Figure 3). Thus, decisions made by architects should be more and more accurate and conscious. They should, moreover, be the knowledge-based decisions in view of the needs of all co-participants of the process.

## 2. Material and Methods

### 2.1. W. Dega Hospital in Poznan (Poland)

The W. Dega orthopaedic and rehabilitation clinical hospital of the Medical University in Poznan [20] provides hospital treatment and general rehabilitation services to patients (Figure 4, Figure 5, Figure 6 and Figure 7).

Orthopaedic diseases are most often the result of a broadly defined injury or degenerative disease of the musculoskeletal system that develops slowly over the years.

The building occupies two wings and is located near the city centre. The hospital accommodates fourteen departments, hospital outpatient facilities, doctor’s offices, and labs. Within the hospital premises, there is a swimming-pool, a modern rehabilitation centre with rehabilitation-supporting technologies and sport facilities. Functionally the hospital is organised into a hospital and an outpatient facility.

### 2.2. Methodology

This study applied a range of research methods selected in view of the assumed research goals. Among the methods used were a review of the available literature and descriptive analyses. National and international scientific publications, databases, reports, and statistical data served as our reference material.

The research included qualitative and quantitative studies developed on the basis of statistical methods and techniques. Its purpose was the diagnosis of spatial needs of users of an orthopaedic and rehabilitation hospital (i.e., patients and staff) and their preferences as to the selected amenities. The following research techniques were used: (1) description, explanation, and interpretation; (2) study of the reference literature, analysis, and criticism; (3) collecting documentation and artefacts, taking photographs, making sketches, etc.; (4) making surveys and using techniques of statistical analysis; (5) correlation; (6) observation; and (7) interviews.

The qualitative approach was chosen as it makes it possible to gain insights about staff and patient perceptions, their spatial preferences, and architectural expectations as well. Qualitative study was carried out based on 119 in-depth interviews developed over one month (April 2022). Participants were doctors (*n* = 12), nurses (*n* = 8), physiotherapists (*n* = 24), and patients (75). All respondents were selected through the current hospital staff and patients, and all of them took part voluntarily in the research. An inductive method was used to answer the following research question:
Personnel perspective:  1. What emotions does work in the orthopaedic ward evoke in you?  2. What are your expectations for hospital space improvements? (Choose from those listed below or enter your own)  - Horizontal circulation routes  - Short distances between premises  - Clear identification system  - Rest and refreshment rooms with direct exit outside  - A social room for a medical staff of all types of specialization  - Cloakroom with lockers  - A kindergarten or creche for the employees’ children  - An option of preparing or buying a warm meal in the workplace  - Attractive location  - Pieces of art in hospital interiors  - Other:….Patient perspective:  1. What emotions does work in the orthopaedic ward evoke in you?  2. What are your expectations for hospital space improvements? (Choose from those listed below or enter your own)  - Horizontal circulation routes  - Short distances between premises  - Clear identification system  - Rest and refreshment rooms with direct exit outside  - A social room for a medical staff of all types of specialization  - Cloakroom with lockers  - A kindergarten or creche for the employees’ children  - An option of preparing or buying a warm meal in workplace  - Attractive location  - Pieces of art in hospital interiors  - Other:….  3. What are additional functions missing in the hospital that are concerned or well-proven in other similar facilities?  - Therapeutic garden  - Outdoor gyms  - Cinema/open air cinema  - A place for work  - Single rooms  - Double rooms  - Accessibility of services, e.g., a store  - Common areas dedicated to social integration  - Restaurant, swimming pool, chapel

By courtesy of the hospital administrator, we were permitted to conduct our qualitative studies in April 2022. We carried out qualitative surveys of the hospital users (staff and one employee) with regard to their perception of the treatment process in the orthopaedic and rehabilitation department. The qualitative studies consisted of conducting prior planned interviews with the medical and administrative staff, based on a set scenario. The interviews intended to find out what the work-facilitating hospital amenities and work-impeding factors were as well as to identify aspects that positively or negatively contributed to patient recovery from the daily perspective of the staff and patients.

The moderator started the interviews by welcoming the participants and presenting herself and the scope of the project. After illustrating the aim of the study and gaining permission for audio-recording, respondents were asked to briefly describe themselves and just after investigating the main research topics proceeded.

Once completed, data were aggregated and analysed.

### 2.3. Qualitative Studies—Results

First, the hospital site visit inspection was made, and then the nursing staff were interviewed.

The site visit started in the admission room. The nurse in charge was first interviewed with regard to the structure and organisation of that part of the hospital: the admission room for planned hospital admissions and registration office are located as separate hospital zones. Wide halls where patients can comfortably wait for their turn, with no impediment to patient and staff traffic are very important too. It was stressed that a queue management system with a ticket printer and a ticket number display at a respective doctor’s office in one larger waiting room space provided with rows of chairs had proven a very good solution. The best working scheme for the patient admission process in the admission room zone was the sequential location of the registration office; next was the nursing interview room, specialised outpatient room, and exit, with an assumption of a one way patient traffic.

The site visit continued at the department of spine orthopaedic surgery and rehabilitation for adults, the organisational structure of which was studied in detail. The main problematic issue at the department proved to be too small or poorly ventilated utility and storage rooms. The head nurse, showing us around the department, took us to a toilet with a bathtub installed on a raised floor level, and pointed out that it should have been replaced with a shower adapted to persons with limited motor abilities: it sometimes happens that patients with limited motor abilities are unable to wash themselves. With this design solution, we are not able to take the patient out of the bathtub as it is very uncomfortable and dangerous. What we should have here is simply a shower head and a shower drain, so we can push a patient on a wheelchair under the shower and wash him/her.

**Figure 5 ijerph-19-15388-f005:**
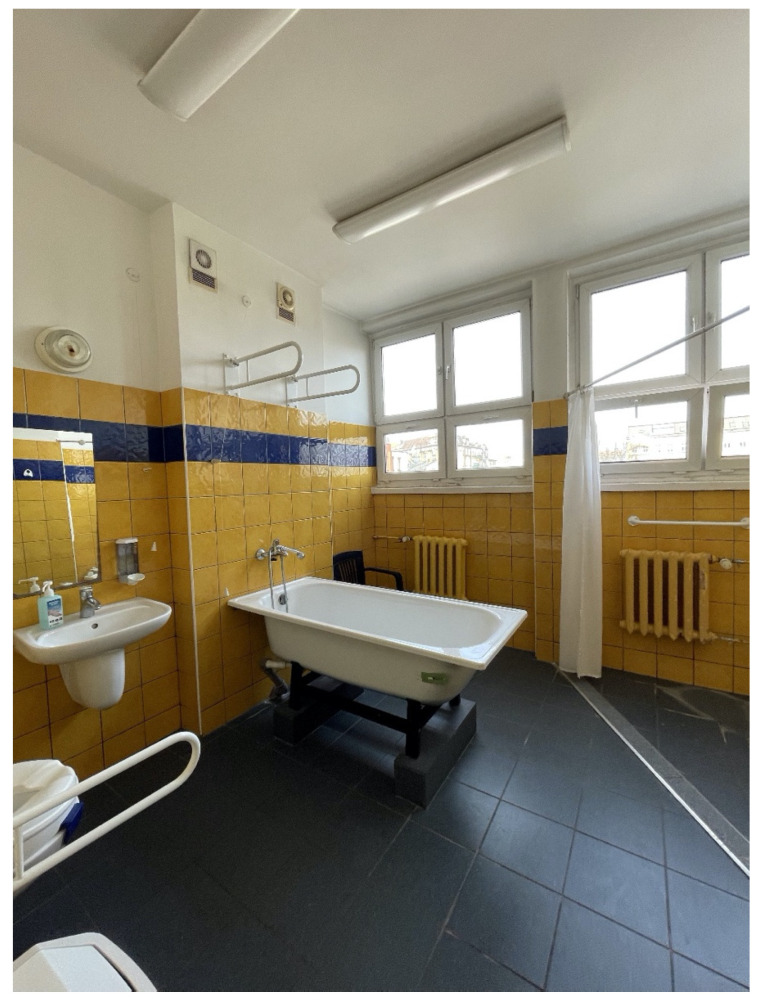
Orthopaedic hospital in Poznan, source: author’s own material.

From the point of view of the nursing staff, a lockable nurse’s station was needed, and not just a counter that is accessible to anyone: in such an organisational system each patient and visitor has direct access to documentation, and even if the documentation is kept in lockers they can be easily opened. Another issue brought to our attention was the sequence of rooms intended for respective stages of inpatient care. The nursing station, followed by the staff rest and refreshment room, treatment room, and finally the wound care unit was assessed as the best organisational scheme. Such a layout of rooms facilitates efficient work organisation and inpatient care.

Regarding the organisation of traffic in the orthopaedic and rehabilitation department, doors proved a significant aspect, to be more precise—the manner of door opening. Both patients and medical staff thought that the safest and the most comfortable solution was the single-wing sliding or tilt and turn doors.

**Figure 6 ijerph-19-15388-f006:**
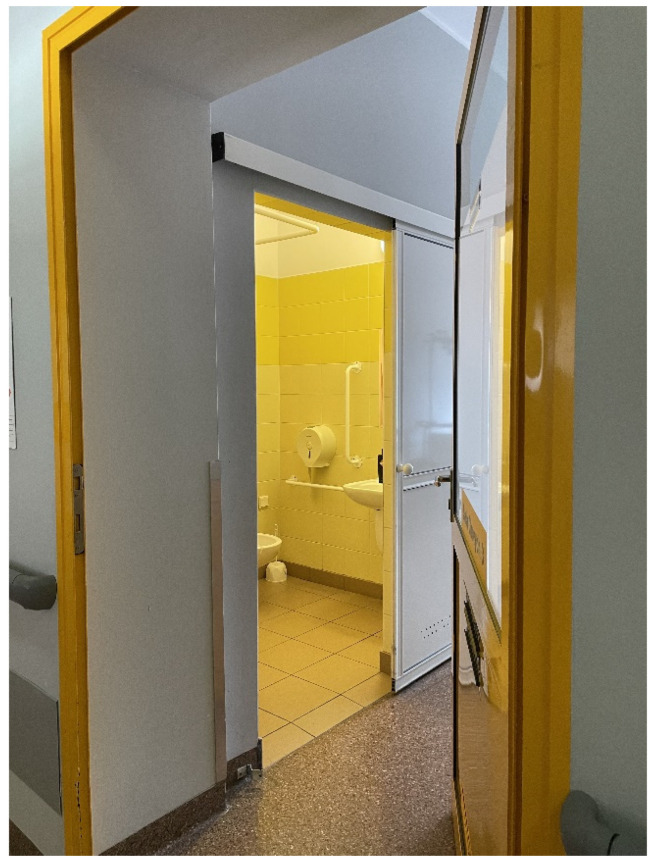
Orthopaedic hospital in Poznan, source: author’s own material.

During the site visit, lighting was another inspected issue. It was found that floor spotlights/light strips were thought to be the best solution because they did not dazzle a person going to the toilet at night or a patient taking medications at night. The bed steering mechanism was another important hospital deign aspect that fell into the area of our research interest. Few people are able to properly adjust a bed with a patient on top if it is provided only with a hydraulic lift. Thus, the importance of a remotely controlled mechanism was brought to our attention.

The operating theatre was the zone we inspected next. In the opinion of the staff, the problematic issue was the space. The large size machines and equipment used to perform the surgeries occupied even the space of the hall. On the other hand, if the machines were to be stored in a dedicated storeroom, their arrangement therein would have to be so designed that each could be freely transported to the operating theatre, with no necessity to shift any other machines aside. As one of the surgeons pointed out: this would require so much space that it seems a waste to design such storerooms.

The department also had a problem with storing baskets with surgical supplies and trays with sterile surgical instruments:

They usually arrive at the operating theatre on a cart, and their excessive spare numbers are usually left in the halls. The operating theatre could do with at least three large storerooms for carts with surgical supplies and instruments, with an option of placing the baskets and trays on one and the same shelf level of a rack system.

After the surgery, the cart with dirty materials is transported down by a lift; however, as was pointed out that the height of the lift space was sometimes too low to accommodate a cart with a pile of stacked dirty materials.

In operating theatres of an orthopaedic and rehabilitation hospital, equipment is not recommended to be directly fixed to a ceiling because surgical procedures are so diverse and concern such a wide range of body parts that all the machines and equipment items must be mobile. Handheld instruments placed on mobile surgical instrument tables is a much better solution, as we were told by one of the surgeons in the operating theatre.

The last visually inspected hospital area was the rehabilitation centre with rehabilitation supporting technologies, a relatively new and well-equipped hospital facility. In the rehabilitation centre all types of rehabilitation equipment for physical therapy can be found, i.e., Neuroforma stations for cognitive and motor rehabilitation with the use of virtual reality technology or Lokomats robotic rehabilitation devices for physiological gait rehabilitation of adults and children.

**Figure 7 ijerph-19-15388-f007:**
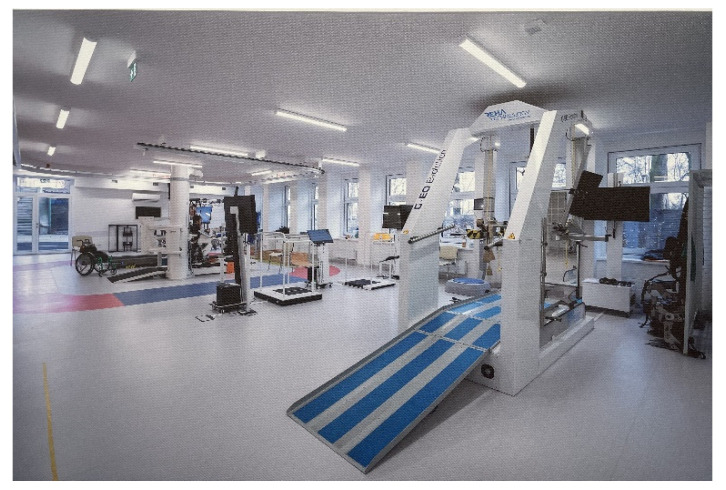
Orthopaedic hospital in Poznan, source: author’s own material.

The rehabilitation room features an open-space design. Among advantages of such a design, we can list easy access of a physiotherapist to all patients; however, mobile medical privacy screens separating equipment and patients were missing; as a result, the camera range could also record the space behind the exercising patient. Noise generated by the machines posed another problematic issue. The need to install acoustic panels was necessary. As one of the physiotherapists stated: despite an open-space arrangement facilitating user integration, it would be an advantage to have individual consultancy rooms accessible directly from the open space rehabilitation area, an equipment storeroom and space where parents or carers can wait for patients. Usually, one rehabilitation therapy session of a child takes 4.5 h, their parents or carers should have a space where they can wait.

The studies were next supplemented with an interview with a 2nd year student of physiotherapy at the Medical University in Poznan, Ms. Maria Tuczyńska, an intern at W. Dega Hospital. The interview, among others, pertained to internal traffic issues. It was found that the general traffic circulation layout was well-organised; however, at some exits there were threshold strips. This posed an obstacle for patients in wheelchairs and those after a knee replacement surgery. Such persons may find it difficult to pass such obstacles or fail to notice them. In general, there are many uneven surfaces in hospitals that pose obstacles mainly to patients in wheelchairs. This was counterbalanced with lift options. As a result, every department can be reached by a lift. The hospital is also provided with two platform stairlifts, one facilitates access of the disabled people to the day ward and the other their exit from the registration office. The lifts are located at the main entrance, near the reception area, and at the entrance to the day ward. As far as the first platform stairlift is concerned, it proves very useful. There is always someone that can assist a disabled person with its operation, whereas regarding the platform stairlift to the day ward, a disabled person in a wheelchair must manage on their own as there is no one that can be asked for help. Another solution to facilitating access of the disabled is wheelchair ramps. In general, staircases are provided with wheelchair ramps, says Maria. However, she has also talked to a volunteer physiotherapist friend who runs physiotherapy programs for people in wheelchairs and in her opinion, from the perspective of a person with limited mobility, these ramps are very steep. Nevertheless, it is good that we can use the option of wheelchair ramps; they are a verified solution and to some extent facilitate access to various hospital areas. When asked about clarity of visual signs, Maria answered that the hospital area was huge, and signs were not always visible at first sight and can be difficult to find. Identification of the building entrance doors is quite complicated as there are no signposts outside. Departments are clearly designated; each has its nameplate displayed over the entrance doors. Hospital external areas such as benches, fountains, and greenery make up additional amenities. The option of reaching them by car from the car park is an additional asset. With the children that need to be hospitalised for longer time periods in mind, the building has been provided with such amenities as a small store, school or a hairdresser.

Maria Tuczyńska stresses that the building is a bit grey and gloomy but there are also some nicely renovated departments or places such as, e.g., a historical staircase with wooden stairs, stucco decorations, and a chandelier. Moving along the hospital, it can be easily observed that halls in the basement level are relatively narrow. This does not pose any excessive nuisance, but as it is also a place where patients arrive, it may be difficult to pass someone in a wheelchair.

### 2.4. Quantitative Research—Results

The above-described qualitative studies were supplemented with respective surveys. A group of anonymous respondents, in this case medical staff and patients, were surveyed in paper and electronic form. One hundred and nineteen respondents took part in the survey, of which forty-four were the employees of the healthcare centre or of an orthopaedic and rehabilitation hospital and seventy-five were the patients. For that purpose, active research method that also actively engages the researcher/expert was used.

The results have been grouped per the surveyed medical staff and patients and can thus be easily compared. The first two questions concerned the sociodemographic data of the respondents, i.e., gender and age. In both groups (medical staff and patients) female respondents dominated—70.5% and 88%, respectively.

The data collected in the two groups showed that over 3/4 of respondents were aged from twenty-one to thirty (Figure 8 and Figure 9). The data collected in the two groups showed a slight difference as regards other age ranges. In the group of patients, none was in the age over 61.

The survey next moved to analyse the emotions invoked by the environment of an orthopaedic and rehabilitation hospital in persons hospitalised or working therein. Three options were envisaged: positive, neutral, and negative. The obtained results clearly show a difference between the feelings of medical staff and patients. A large majority of the respondents had a neutral emotional attitude to the hospital; however, comparing answers provided by the group of hospital employees and patients, it can be noted that the two groups absolutely disagreed. Medical staff assessed their work positively (40.9%) and only 9.1% voiced a negative opinion (Figure 10), whereas only 14.7% of the patients had a positive opinion regarding their stay in hospital and 24% were of a negative opinion (Figure 11).

The next part of the survey included a list of multiple-choice questions on the expectations concerning work or hospitalisation/stay in an orthopaedic and rehabilitation department. Depending on the group of respondents, answers to some questions differed.

For medical staff the most important was an option of preparing or buying a warm meal in a workplace (84.1% of the respondents). This was followed by a rest and refreshment room with a direct exit to outside. A proportion of 70.5% of the respondents selected that option. The sequence of other selected options was as follows: short distance between premises (61.4%), attractive location (56.8%), and cloakrooms with lockers (54.5%). During our site visit, we learnt that the hospital staff highly approved of the locker system, yet they stressed that the number of lockers was insufficient. The respondents also thought that the following were important: horizontal circulation routes (50%), a social room for medical staff of all types of specialisations (50%), and a kindergarten or crèche for the employees’ children (34.1%). Pieces of art in a hospital interior were viewed as the least important. Only 11 persons ticked that option, i.e., 25% (Figure 12).

Patients’ answers also included suggestions as to what they viewed as required in an orthopaedic and rehabilitation hospital (Figure 13). Among the listed items were: an area for walks, high standard of rooms, nice and qualified staff. In the surveyed group of patients, short distances between premises or doctor’s offices were viewed as a top priority (66.7%). Horizontal circulation routes ranked similarly to an attractive location (36% and 34.7%, respectively) and were followed by a clear identification system (33.3%). Like medical staff, the patients also thought the least of art pieces displayed in the hospital interior; nonetheless, this option was selected by 12 respondents (16%).

The patients were asked about additional functions missing in the hospital concerned or well-proven in other similar facilities (Figure 14). An absolute majority of respondents thought that access to medical services was the most important criterion (73.3%). The majority of the respondents preferred single rooms (66.7%) to double rooms (only 8%). Therapeutic gardens (57,3%), outdoor gyms (36%), and common areas dedicated to social integration (34.7%) also ranked high. Of the respondents, 30.7% thought a place for work could be useful and 24% would be happy to have an open air cinema. One person suggested a chapel.

Another question was addressed to the two groups of respondents and referred to a location that might be work-stimulating or recovery-stimulating during the patients’ hospitalisation or stay in an orthopaedic and rehabilitation centre. The obtained results differed depending on the purpose of the respondents’ presence in the facility concerned. Location of a work facility in an urban or suburban area seemed to be the most attractive factor for the employees (Figure 15). Yet, urban and suburban locations were similarly assessed (38.6% of the respondents). Only one in ten of the respondents would choose a facility in a low-urbanised area (22.7%). Opposite opinions were voiced by the patients. Here, a suburban location and a low urbanised location scored similar results (45.3% and 46.7%, respectively), whereas the lowest number of patients were interested in staying in a hospital or healthcare centre in an urban area. Only six persons, i.e., 8% ticked that option (Figure 16).

The last question concerned signposts facilitating patients and staff orientation and circulation in the building. Both groups clearly preferred visual signposts such as maps, schemes, and icon signs. These were followed by horizontal signs. Vertical signs or oral directions were rated at a similar level and viewed as the least legible. The added comments suggested a combination of visual and verbal designations, which seems like quite a good remark (Figure 17 and Figure 18).

## 3. Discussion

A wide range of recommendations on proper design of hospital space have been developed in view of the expectations and potential additional functions. Emotional attitudes of persons working and staying in the orthopaedic department were also analysed.

The obtained results laid the foundations for the design guidelines of an optimum orthopaedic and rehabilitation facility from the perspective of patients and medical staff. The large group of respondents was dominated by women; it may thus be easily inferred that men are not affected with motor dysfunctions to the same degree.

The question whether the respondents had a positive, neutral, or negative emotional attitude to their stay in the orthopaedic and rehabilitation department allowed us to find that the majority felt neutrally about their stay. This means that the treatment does not in any way influence social perception of the condition of the respondents. It is worth noting, however, that negative and positive opinions voiced by the respondents working or staying in the said department rendered opposite results. Concluding, it was mostly patients that expressed a negative rather than a positive opinion about their stay in hospital, whereas among the medical staff, a positive emotional attitude prevailed over a negative one.

The respondents were next asked about their expectations regarding their work or stay in the aforementioned department. Here, only three key issues, selected by the majority of the respondents is presented.

Medical staff thought that an option of preparing or buying a warm meal in a place of work was the most important. As a result, the design guidelines took the account of such a need, and it was decided that a kitchenette should be envisaged in each rest and refreshment room situated in respective departments or the outpatient area.

Another need indicated by a large group of respondents was a rest and refreshment room with a direct exit to outside. This assumption is well-justified in view of the working environment of the medical staff.

Short distances between premises ranked third among the medical staff sub-group of respondents. Taking into account extensive hospital structures and functional connections, the building form design envisaging the most important parts of the hospital in the central part of the facility that is accessible from many different sides seems the best solution. This way, optimum circulation routes between departments and other parts of the hospital can be ensured.

Answers provided by the patients showed that they most appreciated short distances between premises and horizontal circulation routes. It is fully understandable in view of their health problems being the reason of their stay in this type of a medical facility. Moreover, for this group of respondents, a central building body concentrating key hospital functions would also be an optimum building layout.

Attractive location of a hospital was the third preferred option in the patient sub-group of respondents. It is an undeniable fact that patients coming for a surgery or rehabilitation should spend much time doing physical exercises. Attractive hospital location may only facilitate their recovery.

Another question was addressed only to the patients and was intended to find out if there were any additional functions that the hospital concerned was missing or that were well-proven in other similar facilities. It may be concluded from the provided answers that the patients expected availability of services such as a store or a pharmacy and single rooms. Thinking of privacy and the recent COVID-19 pandemic, such expectations are fully reasonable. The third most often selected option was therapeutic gardens intended to support rehabilitation of systemic disorders.

Drawing conclusions from the results obtained regarding location of a hospital, it should be noted that the two groups of respondents provided opposite answers, thus, the answers were predetermined with the purpose why a given respondent stayed in a given healthcare facility. Medical staff clearly preferred a location in an urban or suburban area due to the fact that they had to commute thereto on daily basis and because such a location was close to other facilities. The patients thought that suburban areas or low-urbanised areas offered better location prospects. Such an opinion stems from the specifics of the treatment and rehabilitation. Expecting a long hospital stay, it is no wonder patients cherish natural environment and landscape assets. Such an environment can not only positively affect their physical condition but may also prove conducive to mental repose.

Answers provided to the last question show what hospital signposts were deemed the most legible and easy to understand for the respondents. Here, both the patients and medical staff were in line and highly rated visual signs. This option was followed by horizontal signs (both groups were again concordant in this respect). The above results lead us to conclude that signs that are easily spotted, simple to understand, and lead directly to the destination are most friendly for the patients and medical staff.

The above results provide useful design guidelines and enable us to define specific amenities needed in an orthopaedic and rehabilitation facility by hospitalised patients and working staff. Such guidelines will without doubt facilitate experience and knowledge driven hospital designs up to the satisfaction of medical staff and patients.

## 4. Conclusions

Modern trends tend to shift hospital (including geriatric wards) designs towards a patient-centred design. In particular, they focus on the effect of wellbeing and stress reduction [12,21,22,23,24]. At a very general level of considerations, it may be stated that functional and spatial solutions shall to the widest possible extent ensure the patient’s independence and unassisted living. Hospital space shall be designed with easy orientation of the patients in mind. At the same time, their feeling of dignity and privacy shall be respected at any stage of the treatment. The available field-specific literature offers a wide range of design guidelines that are based on the studies of good professional practices and case studies rather than on legislative requirements.

Forecasts on how the healthcare system and hospital architecture have to transform stress the importance of user participation in the overall process of architectural design of hospitals [25], in this a particular role of patients in the treatment process and in the process of treatment management (patient-centred design and co-design) in the architectural design of healthcare facilities [24,25]. The adopted methodology has confirmed the earlier findings [26,27], namely that it is the role of a designer and researcher to include all the users in the design process of the facilities concerned. If the expectations of the patients’ and medical staff’s quality of the environment is ensured, their comfort and well-being improve. In this area, architecture can play a significant role and predetermine the overall treatment process [28,29,30,31,32,33,34,35].

## Figures and Tables

**Figure 1 ijerph-19-15388-f001:**
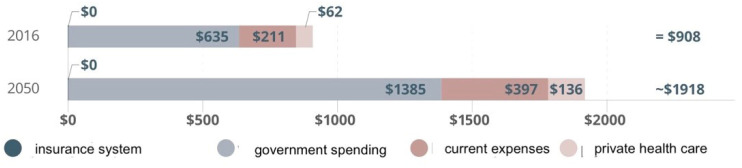
Healthcare spending by type. Prepared by: A. Gawlak, based on: Financing Global Health Database 2018/Financing Global Health Database, http://www.healthdata.org/policy-report/financing-global-health-2018-countries-and-programs-transition, access: 10 August 2019.

**Figure 2 ijerph-19-15388-f002:**
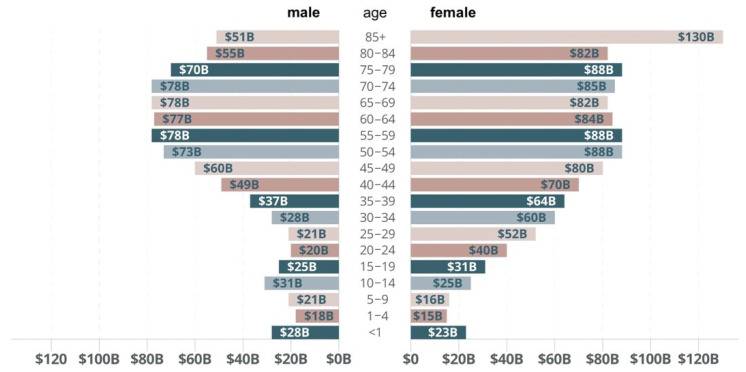
Costs of medical treatment of patients by age and sex [13] by: A. Gawlak.

**Figure 3 ijerph-19-15388-f003:**
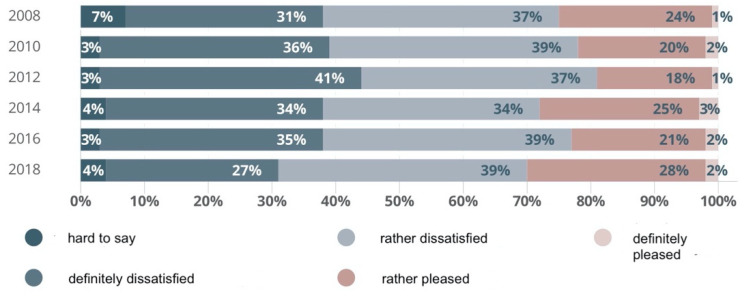
Level of satisfaction from healthcare system in Poland. Prepared by: A. Gawlak, on the basis of the data published by the Polish Supreme Chamber of Control and Centre for Public Opinion Research (Communication no. 89/2018).

**Figure 4 ijerph-19-15388-f004:**
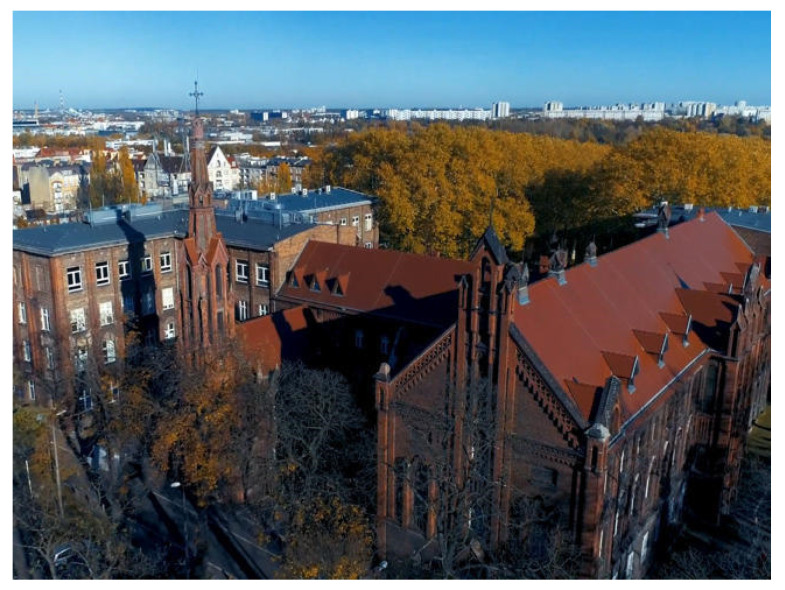
Orthopaedic hospital in Poznan, source: https://www.siepomaga.pl/degapoznan, access: 15 June 2022.

**Figure 8 ijerph-19-15388-f008:**
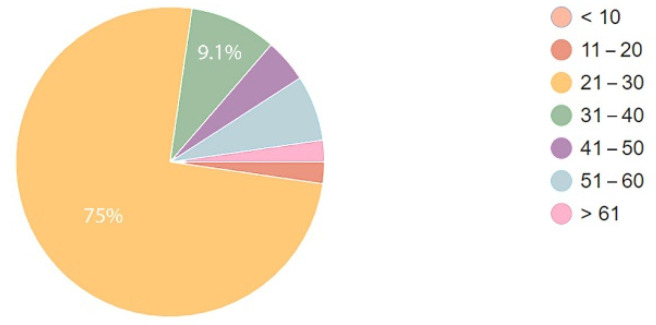
Age of respondent staff.

**Figure 9 ijerph-19-15388-f009:**
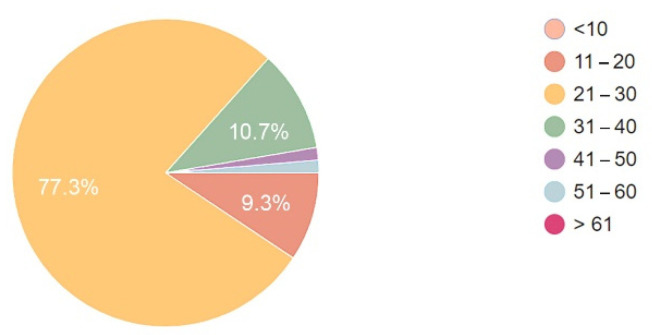
Age of respondent patients.

**Figure 10 ijerph-19-15388-f010:**
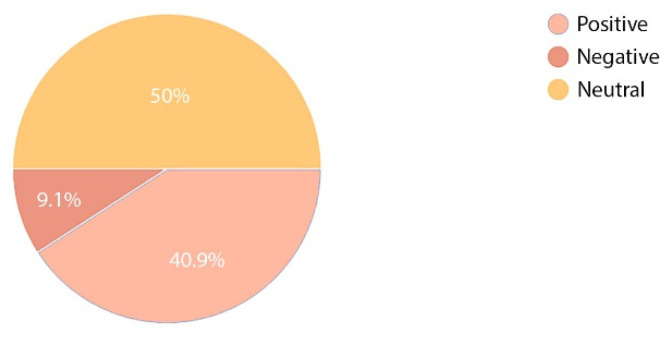
Perception of stay at the hospital—staff.

**Figure 11 ijerph-19-15388-f011:**
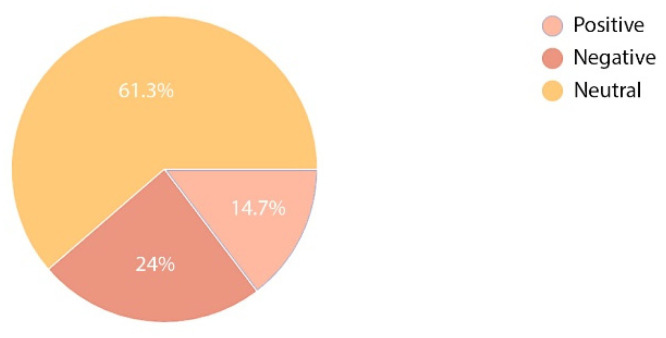
Perception of stay at the hospital—patients.

**Figure 12 ijerph-19-15388-f012:**
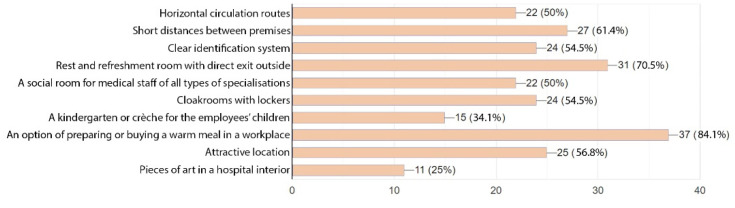
Spatial expectations of staff.

**Figure 13 ijerph-19-15388-f013:**
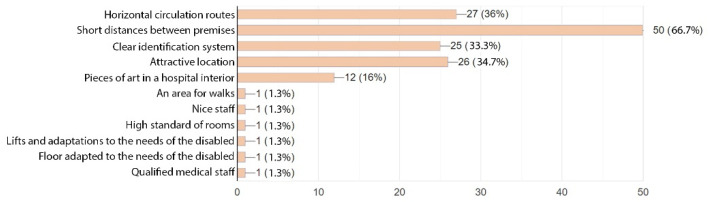
Spatial expectations of patients.

**Figure 14 ijerph-19-15388-f014:**
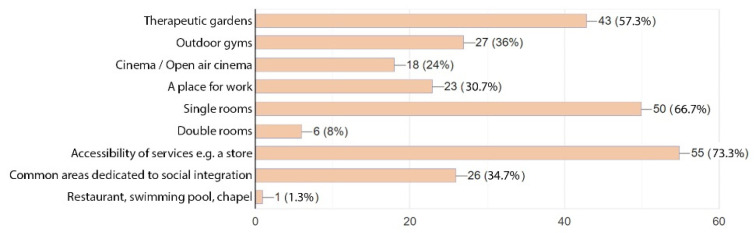
Spatial expectations—alternative amenities—patients.

**Figure 15 ijerph-19-15388-f015:**
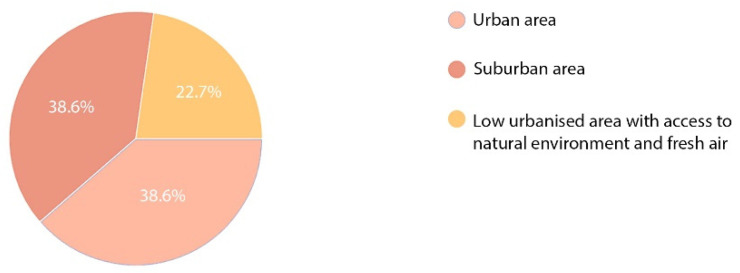
Preferred location of a hospital—staff.

**Figure 16 ijerph-19-15388-f016:**
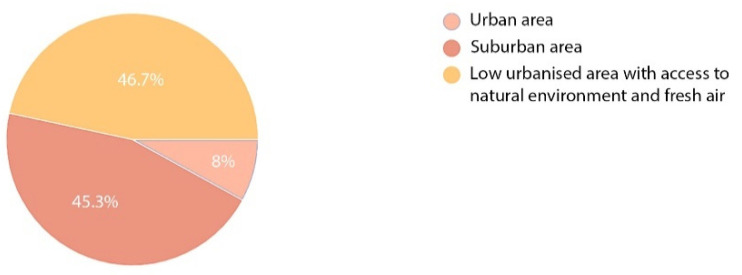
Preferred location of a hospital—patients.

**Figure 17 ijerph-19-15388-f017:**
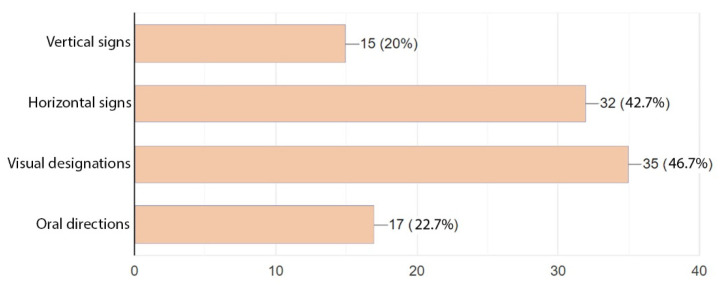
Preferences as to the visual identification system in a hospital—staff.

**Figure 18 ijerph-19-15388-f018:**
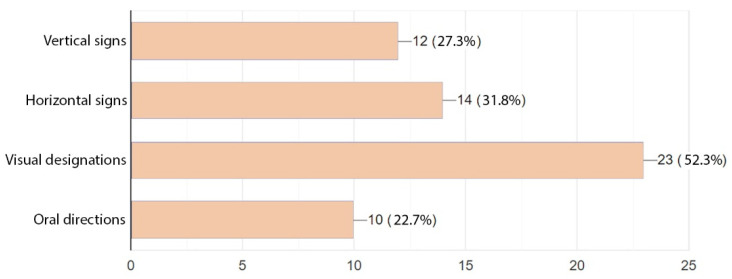
Preferences as to the visual identification system in a hospital—patients.

## Data Availability

The data that support the findings of this study are available from the corresponding author, A.G., upon reasonable request.

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
