# Peer review of "Specific Needs of Patients and Staff Reflected in the Design of an Orthopaedic and Rehabilitation Hospital—Design Recommendations Based on a Case Study (Poland)"

_ijerph, 2022, doi:10.3390/ijerph192215388_

Round 1

Reviewer 1 Report (New Reviewer)

Overall, this study is exciting and valuable as it clarified how a built healthcare facility works, identified what can be improved, and highlighted what needs to be addressed in future hospital building design. Below are areas that can be improved. 

·      Abstract: the first paragraph can confuse readers as it is unclear whether the faculty conducted the research or participated in it. In addition, the result and Conclusion are not cohesive. How does the result link to the Conclusion?

·      Introduction: the introduction needs the most improvement. First, citations are missing in many places. For example, for the sentence starts with "As patients, …" in line 48, could you please add a citation? Also, the four basic aspects of the built environment in line 66 and hospital treatment costs in lines 113-114 need reference support. Second, it is probably unnecessary to include detailed information about journals and researchers, e.g., the Journal of Environmental Psychology in line 75 and a list of researchers in lines 83-84. Moreover, the study was conducted in an orthopaedic hospital, so it is helpful to introduce this type of healthcare facility. How many are there in Poland, Europe, and the world? What is their current status? This introduction will help show the significance and implication of this study. Furthermore, all the figures need to be referred to in the text to facilitate understanding.

·      Purpose of the study: it is unclear what the study's goal is since the background introduction does not identify a clear gap in the field. As the text in lines 179-182, even though the number of the hospital is decreasing, the group of satisfied patients increased. Why is there a need to improve the design of healthcare facilities when the patients are satisfied with the service? 

·      Methods: It is a good idea to apply qualitative and quantitative methods to supplement each other. However, did the 119 respondents participate in both qualitative and quantitative studies? From the result, it seemed that only a few people were interviewed, such as a nurse, a staff, and an intern (Maria) [Do the authors have the agreement of the interviewee to show her name?], but it was not clear in the method description. Another concern with the method is that the authors mentioned the need for healthcare facilities is growing due to the increasing number of older adults. However, most participants in the study were between 21-30. Therefore, a discussion of the limitation of the study is necessary. 

·      Result and Conclusion: similar to the introduction, I hope the authors can elaborate on the implication of this study, e.g., how this study can improve future design practice. 

Author Response

Dear Reviewer,

thank you for timely and insightful feedback on the manuscript. We appreciate all yours suggestions. According to the reviews, we have taken the opportunity to revise our manuscript and to clarify our aims and provide further explanations where it was necessary, as well. We have provided a more complete description of the method, goals and the framework used in the search strategy.

Below you will find our response comments (in blue).

Thank you for encouraging all improvements.

Yours Sincerely,

 Agata Gawlak (Corresponding Author)

Reviewer 2 Report (New Reviewer)

General opinion:

The topic of the paper is of great importance for user-centered design in hospitals that serve patients, and the obtained results can contribute to improving the well-being of patients in hospital interiors. As such it is worthy of further consideration and publication.

Although the manuscript is scientifically based, it is written in non-scientific first-person vocabulary, is not well structured, and is inconsistent in most parts:

- The introduction is too long and quoted in an inappropriate way,

- The results are not well structured,

- Discussion and Conclusion are mixed,

- Literature is not written according to guidelines (missed DOI, web pages not found, some citations missing author name, etc.)

- The paper is not written in accordance with the instructions to the authors (e.g. Pictures are not cited in the text)

-The objective is not clearly stated, so it is suggested to state it at the end of the 1st chapter.

It should be noted that the work deals with research on/with patients, so in the article, it is desirable to better explain why the approval of the ethics committee was not obtained and to indicate more clearly how the approval of the respondents to participate in the research was requested.

Specific comments:

Line 7: Affiliation – Please add the e-mail address of the second author

Lines 9-29: Abstract could be to be more concise and shorter. There is no need to separate paragraphs with the enter sign. Remove headings (methods, discussion, conclusion), and fit them substantively as part of a sentence

Lines 30: keywords: put the main keyword first (e.g. hospital design) the word "participation" can even be last

Line 35-40: the sentence/paragraph does not make sense if it starts with the words „

Please reform the opening paragraph. The first sentence must not begin with the words "On the one hand...". The whole section (lines 35-40) is a bit confusing. The first sentence is too long and could be written in several shorter clear sentences with references.

Line 36: …over 65 years old increases

Line 51: check the initials of the author or mention the other authors (et al). E.g.: „Urlich et al [4] report a ….“

Line 66: which literature? Please name in references.

Line 66: number 4 is better to be written as a word, not a number. The same in line 71

Line 71: The above four criteria…

Line 72: delete „such“

Lines 73-78. shorten the sentence, there is no need to explain and refer in this way

Line 80: note the order of the references (numbers)

Line 80-81: Correct the beginning of the sentence

Lines 83-84: correct according to the guidelines – delete mentioned names of the authors, shorten the sentence

Line 90: Please refer to the research you mention (by a number in bracket)

Line 99-100: Please refer to a reference to confirm this claim.

Lines 104-106: Please refer to a reference to confirm this claim.

Lines 113-114: Please refer to a reference to confirm this claim.

Lines 119-121: Please refer to a reference to confirm this claim.

Lines 121-123: Please refer to a reference to confirm this claim.

Lines 125-127: use the correct reference (Source). Adapted by A.G. Check the copyrights

Lines 128-129: the same as previous. Check the copyrights

Line 135: Please check the reference – when is it accessed, is there a website etc.

Line 137: Please refer to a reference (referred to as a number in References)

Line 147: the same as line 135

Line 148: Please refer to a reference (referred to as a number in References)

Line 159: the reference [14] can’t be reached

Line 164-165: Please refer to a reference to confirm this claim (referred to as a number in References)

Line 169: the reference [15] is not written according to the guidelines. Please correct.

Line 173: the reference [14] can’t be reached

Lines 33-188: the entire chapter should be shortened and written in the style of scientific language, without using so many conjunctions and adverbs. the advice is to write in the third neutral person and be sure to check all the references listed in this chapter in accordance with the author's guidelines - many of them are not listed correctly (missing DOI, wrong order of initials, no author, not stated when the article was accessed, etc.)

189: if the Chapter is Introduction, please rename Chapter 2 as Materials and Methods

192: Please provide a link to the hospital and refer to it as the reference

199: correct according to the guidelines

202: Extract the work objectively and move it to chapter 1

212-219: Please describe methods more structured

243: It is not clear do the patients work in the department or does the question refer to the observation of the work of the employees.

257: Please explain what the number 415 means.

269-272: please explain better why didn’t you use Ethical Board consent and how did you manage to enter the hospital with the permit to conduct research on/with patients

285-320: please describe the whole Chapter shorter and clearer. Organize the text into subsections according to the results obtained. Use scientific language, please.

Check the citation of all displayed images in the text (none are in this text). The names (descriptions) of the Figures should be different.

406-491: The same comment as for the previous Chapter (2.2.) All figures should be mentioned and clearly described in the text. Divide results into Subchapters.

424: there is no table in the text, please correct

492 and 561: Please check the Titles of the Chapters. The conclusion should follow the Discussion. There should be no references in the Conclusion section – it must clearly state brief conclusions and potential directions for further research or improvement.

505: be careful while concluding in the Discussion section. Correct it, please.

The discussion of the results will be clearer if the author could prepare a ground plan (scheme) of the hospital and display the mentioned locations/spots

561-580: make sure that this Chapter is finalized as Conclusion, not Discussion.

596-648: All references must be written according to the guidelines. Please, check DOI numbers (e.g. ref. 1, 5, 16, 19, 21, 27, 28, etc), properly written names and initials of authors (e.g. 6,11,,etc), in some references the names of the authors are missing (e.g. 7, 8, 10, etc), some websites can’t be reached (e.g. 7,8,12,13, 14, etc)

It should be noted that the topic of the article is very interesting, inspiring, and worthy of publication and further research. The comments presented in the review are of a stimulating nature in order to improve the scientific way of presenting the topic and the clarity and brevity of the text.

In addition to the results, it would be desirable to compare the responses of employees and patients. Please consider the above-mentioned comments and please resubmit the corrected article for review.

Thank you.

Author Response

Dear Reviewer,

thank you for timely and insightful feedback on the manuscript. We appreciate all yours suggestions. According to the reviews, we have taken the opportunity to revise our manuscript and to clarify our aims and provide further explanations where it was necessary, as well. We have provided a more complete description of the method, goals and the framework used in the search strategy.

Below you will find our response comments (in blue).

Thank you for encouraging all improvements.

Yours Sincerely,

Agata Gawlak (Corresponding Author)

This manuscript is a resubmission of an earlier submission. The following is a list of the peer review reports and author responses from that submission.

Round 1

Reviewer 1 Report

The work did not investigate the value of participatory design as stated, as it did not engage with that process. It was a survey of hospital staff and patients about their impressions of a very specific facility. It would be better to be clear about this.

The research questions were unclear as was the methodology. These  should be carefully rewritten in one section [see comments in the paper].

The introduction was overly long and not precisely structured to illustrate what the state of knowledge on hospital design with user input is right now, nor does it identify the gap in knowledge that you intended to address with your research.

The data was poorly analysed. It was unclear why you collected age and gender data, or why you presented it, as it was not cross-referenced to answers to identify if there were differences in gender or age groups. There was no explanation as to why the questions were structured in the way they were - was it drawing on previous research? If not, what was it based on?

The qualitative survey and interviews were not precisely described as to what was to be achieved or how they were organised. The results were not analysed systematically, nor used in comparison to the quantitative data.

The resulting conclusions suffered from the lack of coherent research question, clear methodology, and systematic analysis. Any conclusions should have referred back to the original literature review to reflect on how this compared to pre-existing knowledge and what it had to offer that was new to the field.

There are very many citations missing.

Author Response

Dear Reviewer,

thank you for timely and insightful feedback on the manuscript. We appreciate all yours suggestions. According to the reviews, we have taken the opportunity to revise our manuscript and to clarify our aims and provide further explanations where it was necessary, as well. We have provided a more complete description of the method, goals and the framework used in the search strategy.

Below you will find our response comments, suggestions or concerns.

Thank you for encouraging all improvements.

Yours Sincerely,

Agata Gawlak

Reviewer 2 Report

This reviewer did not find a clear statement that defines the main objective of this study. He supposes the authors wanted to do a critical review of the healthcare facilities design process. As the objective is unclear, it is impossible to judge any merit.

Author Response

Dear Reviewer,

thank you for timely and insightful feedback on the manuscript. We appreciate your comment. According to the other reviews, we have taken the opportunity to revise our manuscript and to clarify our aims and provide further explanations where it was necessary, as well. We have provided a more complete description of the method, goals and the framework used in the search strategy. We hope that it meets yours expectations and the high level of IERPH as well.

Yours Sincerely,

Agata Gawlak

Round 2

Reviewer 2 Report

The authors, once again, did not present the objective of their research. Their text looks more like an experiment report. They never stated their objective. What do they want to display with this experiment? Which is the desired conclusion? The authors displayed some of the contexts of this research but never stated clearly what they wanted with it.

The text organization is poor. It is not scientifically sound. We cannot judge if the research design and references were appropriate as the authors did not display the research goals.